# DAAM mediates the assembly of long-lived, treadmilling stress fibers in collectively migrating epithelial cells in *Drosophila*

Kristin M Sherrard[1], Maureen Cetera[2†], Sally Horne-Badovinac[1,2]*

[1]Department of Molecular Genetics and Cell Biology, The University of Chicago, Chicago, United States; [2]Committee on Development, Regeneration, and Stem Cell Biology, The University of Chicago, Chicago, United States

*For correspondence: shorne@uchicago.edu

Present address: †Department of Genetics, Cell Biology and Development, University of Minnesota, Minneapolis, United States

Competing interest: The authors declare that no competing interests exist.

**Abstract** Stress fibers (SFs) are actomyosin bundles commonly found in individually migrating cells in culture. However, whether and how cells use SFs to migrate in vivo or collectively is largely unknown. Studying the collective migration of the follicular epithelial cells in *Drosophila*, we found that the SFs in these cells show a novel treadmilling behavior that allows them to persist as the cells migrate over multiple cell lengths. Treadmilling SFs grow at their fronts by adding new integrin-based adhesions and actomyosin segments over time. This causes the SFs to have many internal adhesions along their lengths, instead of adhesions only at the ends. The front-forming adhesions remain stationary relative to the substrate and typically disassemble as the cell rear approaches. By contrast, a different type of adhesion forms at the SF's terminus that slides with the cell's trailing edge as the actomyosin ahead of it shortens. We further show that SF treadmilling depends on cell movement and identify a developmental switch in the formins that mediate SF assembly, with Dishevelled-associated activator of morphogenesis acting during migratory stages and Diaphanous acting during postmigratory stages. We propose that treadmilling SFs keep each cell on a linear trajectory, thereby promoting the collective motility required for epithelial migration.

## Editor's evaluation

Sherrard and colleagues present one of the first examples of in vivo analysis of stress fibers, adhesive structures that are important for cell migration. While stress fibers have been widely studied in cell culture models the relation between these studies and what is actually seen in intact whole animals remains to be established. Using quantitative live imaging and the vast toolkit of genetic tools available in the fly, they characterize a novel treadmilling behavior of stress fibers, providing valuable information about the mechanisms of cell migration and how the collective movement of cells is regulated in vivo.

## Introduction

Migrating cells rely on dynamic networks of filamentous actin (F-actin) for their motility. For cells migrating on two-dimensional substrates, these include the branched networks that underly lamellipodial protrusions (*Ridley, 2011*) and the stress fibers (SFs) that mediate much of the cell's interaction with the extracellular matrix (ECM). SFs are prominent F-actin bundles that have been categorized into different types based on their origin, subcellular location, and how they interact with nonmuscle myosin II and integrin-based focal adhesions (*Burridge and Guilluy, 2016*; *Burridge and Wittchen,*

**Figure 1.** Introduction to the stress fibers (SFs) in collectively migrating follicle cells. (**A**) Composite images of an egg chamber (pseudocolored), transverse section above and 3D cutaway view below. Curved arrow shows the rotational migration of the follicle cells as they crawl along the basement membrane extracellular matrix (ECM) (drawn as a line in upper image, from a confocal section of Collagen-IV-GFP in lower image). (**B**) Image of the basal surface of the follicular epithelium. Each cell has a leading-edge protrusion (yellow) and a parallel array of SFs (orange) oriented in the direction of migration. Experiments performed at stage 7. Gray arrows show migration direction. Scale bars 10 μm.

*2013*; *Naumanen et al., 2008*; *Tojkander et al., 2012*; *Vallenius, 2013*). The most studied SF types have myosin along their lengths and large focal adhesions at either end; these include ventral SFs (*Hotulainen and Lappalainen, 2006*; *Tojkander et al., 2015*) and the recently defined cortical SFs (*Lehtimäki et al., 2021*). SFs play key roles in focal adhesion maturation, defining the cell's front–rear axis, retraction of the trailing edge, and sensing the mechanical properties of the ECM (*Lehtimäki et al., 2017*; *Livne and Geiger, 2016*; *Schwartz, 2010*). However, our current knowledge of SFs in migrating cells comes almost entirely from studies of individual cells on nonnative substrates. Whether and how cells use SFs to migrate in vivo or as part of a collective is largely unknown.

In this study, we use an in vivo system, the *Drosophila* egg chamber, to probe SF dynamics in collectively migrating epithelial cells (*Cetera and Horne-Badovinac, 2015*; *Horne-Badovinac and Bilder, 2005*). An egg chamber is an ovarian follicle that will give rise to one egg. It has a central cluster of germ cells surrounded by a somatic follicular epithelium; a basement membrane ECM encapsulates the entire structure (*Figure 1A*). During early stages of oogenesis, the basal surfaces of the follicle cells collectively migrate along the basement membrane (*Cetera et al., 2014*; *Lewellyn et al., 2013*). This causes the entire egg chamber to rotate within the basement membrane, which itself remains stationary (*Haigo and Bilder, 2011*). During this migration, each follicle cell has actin-based protrusions at its leading edge and a parallel array of SFs across its basal surface that are oriented in the direction of tissue movement (*Cetera et al., 2014*; *Gutzeit, 1991*; *Figure 1B*). At later stages when the follicle cells have stopped migrating, the density of SFs across the basal surface increases (*Delon and Brown, 2009*; *Gutzeit, 1991*; *Gutzeit, 1990*), and their contractile activity helps to create the elongated shape of the egg (*Cerqueira Campos et al., 2020*; *He et al., 2010*).

Here, we show that the SFs in the follicle cells have many internal adhesions along their lengths, in addition to adhesions at the ends. We further show that these SFs undergo a novel treadmilling behavior that allows them to persist as the cell migrates over more than one cell length. Treadmilling SFs grow at their fronts by adding new adhesions and actomyosin segments over time. These

front-forming adhesions remain stationary relative to the substrate, transition to being internal adhesions, and typically disassemble as the cell rear approaches. By contrast, a different type of adhesion forms at the SF terminus that appears to slide with the cell's trailing edge as the actomyosin segment ahead of it shortens. Blocking migration causes the internal adhesions to disappear and the treadmilling behavior to stop, which shows that the modular SF architecture depends on cell movement. We further identify a developmental switch in the formins required for SF assembly, with Dishevelled-associated activator of morphogenesis (DAAM) contributing to treadmilling SFs during migratory stages and Diaphanous (Dia) contributing to the more canonical SFs that form once migration has ceased. We propose that treadmilling SFs ensure that each epithelial cell maintains a linear trajectory and thereby promote the highly orchestrated collective motility required for tissue-scale movement.

## Results

### Migrating follicle cells have long-lived, treadmilling SFs

To visualize SF dynamics in living follicle cells, we used Green Fluorescent Protein (GFP)-tagged and mCherry (mCh)-tagged versions of the regulatory light chain of myosin II (MRLC, Spaghetti Squash in *Drosophila*) (*Figure 2A*) and near total internal reflection fluorescence (near-TIRF) microscopy (*Figure 2—video 1*; *Figure 2—video 2*). Live actin labels are less effective markers because they strongly label leading-edge protrusions, which obscures the SF tips; they can also disrupt F-actin organization (*Figure 2—figure supplement 1A–D*). This imaging strategy allowed us to watch SFs appear and disappear over time as the follicle cells migrated. Appearance of a new SF is marked by a rapid coalescence of MRLC (*Figure 2B*). SFs disappear either by fading along their lengths or by contracting rapidly from the rear (*Figure 2C*).

We measured SF lifetimes by identifying individual SFs at the midpoint of a 60- min video and tracking their behavior over the full imaging period (*Figure 2D*). Only actomyosin fibers that spanned at least half the cell's length were tracked. Through this assay, we determined that these SFs have a half-life of 34 min. This is clearly an underestimate, however, as only 60% of SFs both appeared and disappeared during the 60- min window. For 32% of SFs, we saw either their appearance or disappearance but not both. The remaining 8% persisted for the entire 60 min. We then used the same dataset to analyze the relationship between SF lifetimes and the distance traveled by the cell. The follicle cells migrated a mean distance of 16.8 µm in this assay, corresponding to a mean of 2.2 cell lengths. Strikingly, 62% of SFs persisted for longer than the time required for the cell to travel at least one cell length (*Figure 2E*). Given that SFs are attached to an immobile substrate by integrin-based adhesions, we reasoned that the entire actomyosin fiber must be treadmilling (i.e., growing at the front and shrinking at the back). Indeed, new actomyosin segments are added to the front of existing SFs (*Figure 2F–H*, *Figure 2—figure supplement 2A–C*, and *Figure 2—video 2*), which explains their persistence. Hereafter we refer to these long-lived SFs as 'treadmilling stress fibers'.

### Treadmilling SFs have many adhesions along their lengths

To understand the treadmilling behavior of the SFs, we examined their associated integrin-based adhesions. Previous descriptions of these adhesions in follicle cells primarily focused on later stages of egg chamber development (*Delon and Brown, 2009*), after the follicle cells have ceased migrating. At these later stages, the SFs are increased in density across the basal surface, and they exhibit the canonical organization with large focal adhesions at their ends (*Figure 3A–D, G*). By contrast, the SFs present during migratory stages have many smaller adhesions along their lengths (*Figure 3A, G*, *Figure 3—figure supplement 1*; *Cetera et al., 2014*), with an average of 5.5 adhesions per fiber. This phenomenon is easiest to see when Paxillin-GFP is overexpressed (UAS-Pax-GFP) (*Figure 3A*); however, we also see multiple adhesions with endogenous GFP tags on Paxillin, Talin, and the βPS-integrin subunit, Myospheroid (*Figure 3B–D*). Thus, the SFs in migrating follicle cells have internal adhesions, in addition to adhesions at their ends.

### Adhesions are added to the front and removed near the back of treadmilling SFs

To determine how the internal adhesions relate to the treadmilling behavior we saw with MRLC, we imaged epithelia in which MRLC was expressed in all cells and UAS-Pax-GFP was expressed in

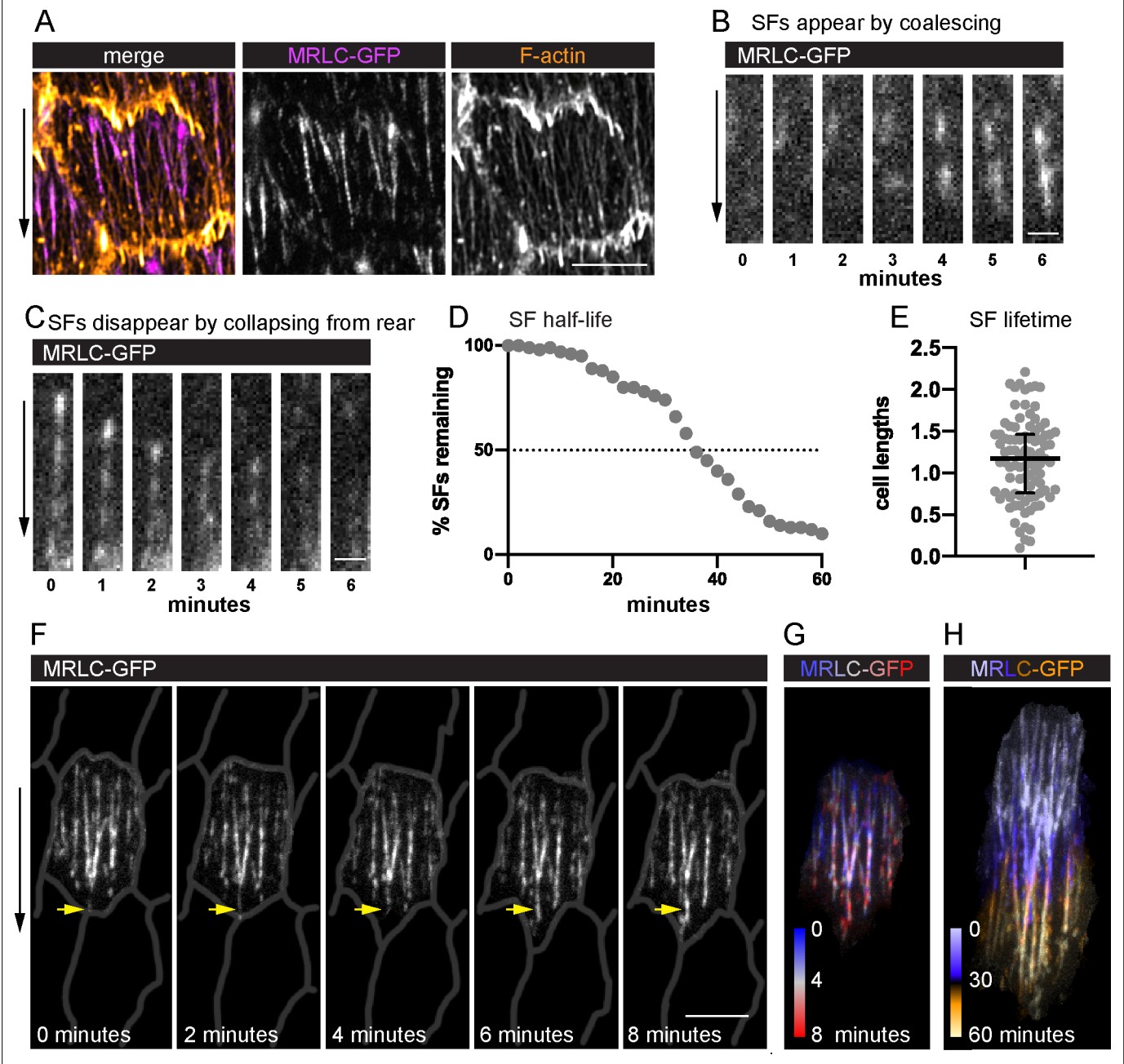

**Figure 2.** Migrating follicle cells have long-lived, treadmilling stress fibers (SFs). (**A**) Image of one cell showing that myosin regulatory light chain (MRLC-GFP) labels SFs, but not leading-edge protrusions. Image shown is representative of five egg chambers. See *Figure 2—figure supplement 1* for comparison of live actin labels. Still images from videos showing an SF (**B**) forming by MRLC-GFP coalescence (*n* = 64 SF appearances), and (**C**) disappearing by collapsing from the rear (*n* = 75 SF disappearances). Quantification of SF lifetimes. (**D**) Half-life measurement in real time. (**E**) Lifetimes as a function of how long it took the cell to migrate one cell length. *n* = 91 SFs from 23 cells in 3 egg chambers. Bars in (**E**) show median and interquartile ranges. (**F**) Still images from a video of an optically isolated cell showing an SF tip growing as the cell migrates (arrow). Cell outlines are drawn from CellMask membrane label. See *Figure 2—video 1*; *Figure 2—video 2*, and *Figure 2—figure supplement 2* for images of the original membrane label and an optically isolated single SF. (**G, H**) Temporal projections of SFs from the cell in (**F**). (**G**) Shows the same period as (**F**) at 20- s intervals. (**H**) Shows the period required for the cell to migrate ~1 cell length. Experiments performed at stage 7. Black arrows show migration direction. Scale bars: 5 µm (**A, F**), 1 µm (**B, C**).

The online version of this article includes the following video and figure supplement(s) for figure 2:

**Source data 1.** Excel file containing the numeric data used to generate the graphs in *Figure 2*.

**Figure supplement 1.** Treadmilling stress fibers (SFs) marked with live F-actin labels.

*Figure 2 continued on next page*

*Figure 2 continued*

**Figure supplement 2.** Treadmilling of an optically isolated stress fiber (SF).

**Figure 2—video 1.** Time-lapse video of a field of follicle cells with plasma membranes labeled with CellMask and stress fibers (SFs) labeled with myosin regulatory light chain (MRLC)-GFP.

https://elifesciences.org/articles/72881/figures#fig2video1

**Figure 2—video 2.** Time-lapse video of an optically isolated follicle cell with stress fibers (SFs) labeled with myosin regulatory light chain (MRLC)-GFP.

https://elifesciences.org/articles/72881/figures#fig2video2

a subset of cells. We then analyzed the dynamics of the adhesions associated with individual SFs using a space–time plot called a kymograph (*Figure 4A, B*). Briefly, we used the MRLC signal to optically isolate one SF from a video (*Figure 4—video 1*), created a one-pixel wide projection from each cropped video fame, and then arranged the projections horizontally to display changes in SF structure over time in a two-dimensional image. This analysis showed that new adhesions and new segments of actomyosin are continuously added to the front of a SF over time (*Figure 4A*). A front-forming adhesion typically first appears near the cell's leading edge and grows in brightness as an actomyosin segment coalesces behind it to link the new adhesion to an existing SF (*Figure 4C*). New adhesions are also pulled slightly rearward, consistent with their maturing under tension. The front-forming adhesions then remain stationary relative to the basement membrane substrate.

We noted that a different type of adhesion forms at the back of the cell. For 82% of the SFs analyzed, the final adhesion at the SF terminus appears to slide along the substrate at the same speed that the cell migrates (*Figure 4A, D*). Sliding adhesions can arise from stationary adhesions; however, 80% arise de novo near the cell's trailing edge. Sliding adhesions are remarkably long lived, persisting for a mean of 34 min, which corresponds to the time required for a cell to migrate 1.2 cell lengths. By contrast, stationary adhesions persist for only a mean of 13 min (*Figure 4E*). Stationary adhesions typically disassemble before they reach the cell's trailing edge. However, ~20% of stationary adhesions are subsumed by sliding adhesions (*Figure 4D* and *Figure 4—video 2*). We observed that 74% of sliding adhesions merged with at least one stationary adhesion, which may account for their long lifetimes.

In summary, a treadmilling SF grows at its front by incorporating new adhesions and new actomyosin segments over time. These front-forming adhesions remain stationary relative to the substrate and typically disassemble as the cell rear approaches. By contrast, a sliding adhesion forms at the SF terminus and appears to move with the cell's trailing edge as the actomyosin segment in front of it shortens (*Figure 4F*).

## The modular architecture and treadmilling of the SFs depend on cell migration

Because new adhesions are added to the front of a treadmilling SF as the cell's leading-edge advances, it seemed likely that the treadmilling behavior would depend on cell movement. We blocked follicle cell migration by employing two methods that eliminate leading-edge F-actin networks in these cells, inhibition of Arp2/3 by CK-666, and depletion of the Scar/WAVE complex with RNAi against Abelson interacting protein (*Abi RNAi*). Analysis of fixed samples revealed that both treatments cause SFs to adopt a canonical architecture, in which the adhesions become concentrated at the ends (*Figure 5A*).

We then used live imaging to explore how this structural transition occurs. Following addition of CK-666, the follicle cells gradually come to a stop. The internal adhesions disappear over time as the end adhesions grow and myosin becomes concentrated in the center (*Figure 5B* and *Figure 5—video 1*). Once this transition is complete, individual SFs often contract from both ends toward the center and disappear; new adhesions are never added to their ends (*Figure 5C*).

The transition from a modular to a canonical SF architecture could be due to loss of cell migration; however, it could also be due to loss of some other activity that depends on the Scar/WAVE complex. To distinguish between these possibilities, we examined SFs in clones of cells expressing RNAi against the Abi or Sra1 components of the Scar/WAVE complex. These mosaic epithelia retain the ability to migrate because the RNAi-expressing cells are carried along by their wild-type neighbors (*Cetera et al., 2014*). Importantly, the SFs within the RNAi-expressing clones show the same modular architecture and treadmilling behavior as wild-type SFs (*Figure 5D, E*). We therefore conclude that it is

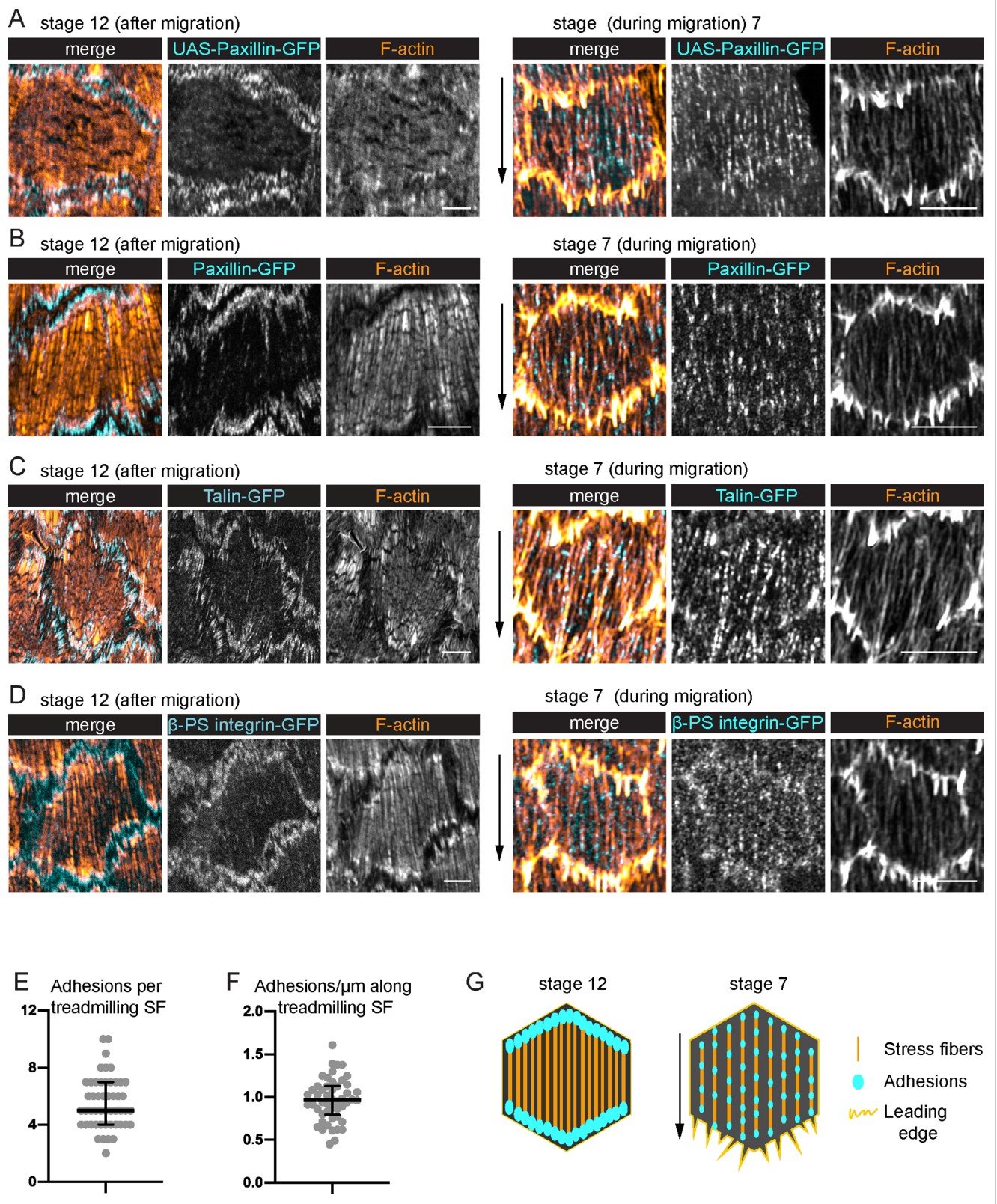

**Figure 3.** Treadmilling stress fibers (SFs) have many adhesions along their lengths. (**A–D**) Images of SFs in single cells with adhesions labeled. After migration stops, there are large adhesions at the ends of the SFs. During migration, there are many smaller adhesions along their lengths. (**A**) UAS-Paxillin-GFP (images shown representative of 4 stage 12 and 3 stage 7). (**B**) Paxillin-GFP (*n* = 4, 5). (**C**) Talin-GFP (*n* = 3, 3). (**D**) βPS-integrin-GFP (*n* = 4, 3). (**B–D**) The GFP on the indicated protein is a functional, endogenous tag. (**E, F**) Quantification of the adhesions associated with individual SFs using

*Figure 3 continued on next page*

*Figure 3 continued*

Paxillin-GFP. (**E**) Number. (**F**) Linear density. *n* = 277 adhesions from 11 cells in five egg chambers. Bars show medians and interquartile ranges. See *Figure 3—figure supplement 1* for method for counting adhesions. (**G**) Illustration of SF structure in postmigratory and migratory cells. Black arrows show migration direction. Scale bars: 5 μm.

The online version of this article includes the following figure supplement(s) for figure 3:

**Source data 1.** Excel file containing the numeric data used to generate the graphs in *Figure 3*.

**Figure supplement 1.** Method for quantifying number of adhesions along a stress fiber (SF).

migration itself, not the activity of the Scar/WAVE complex, that is necessary to generate treadmilling SFs.

## DAAM mediates the assembly of treadmilling SFs

To better understand how treadmilling SFs are built, we sought to identify the source of their F-actin. We first considered that branched F-actin networks flowing back from leading-edge lamellipodia could be incorporated into them. However, elimination of leading-edge F-actin networks through depletion of the Scar/WAVE complex did not reduce F-actin levels in the SFs (*Figure 6A, D*, *Figure 6—figure supplement 1A, C*). This is also true for cells depleted of Enabled, which builds leading-edge filopodia (*Figure 6—figure supplement 1B, C*; *Cetera et al., 2014*). Hence, the leading edge is not a major source of F-actin for these SFs.

We next asked if formins are required, as these proteins assemble unbranched F-actin and are often associated with SF formation (*Kühn and Geyer, 2014*; *Valencia and Quinlan, 2021*). We performed an RNAi-based screen of *Drosophila* formins and found that depleting DAAM reduces F-actin levels in the SFs by ~30% (*Figure 6—figure supplement 1D*, *Figure 6B, D*). We confirmed this result by showing that two null alleles of *DAAM* similarly reduce F-actin in the SFs without having obvious effects on other F-actin populations (*Figure 6B, D, E*), and that an activated form of DAAM increases F-actin in the SFs (*Figure 6C, D*). RNAi against other formins had no effect on the SFs, nor did codepleting formins or other F-actin assembly factors with DAAM (*Figure 6—figure supplement 1D, E*). From these data, we conclude that DAAM is a key contributor to treadmilling SF assembly. It is important to note, however, that we do not know that all the formin RNAi transgenes we screened are functional, so other formins may work with DAAM in this context.

Finally, we asked how DAAM contributes to treadmilling SFs. Using a line in which DAAM is endogenously tagged with GFP (DAAM-GFP) (*Molnár et al., 2014*), we found that DAAM localizes largely uniformly within the cortex with no obvious enrichment on SFs or adhesions (*Figure 7A, B*). DAAM-depleted cells have the same density of SFs across their basal surfaces as control cells (*Figure 7C, D*); each one simply has reduced levels of F-actin, myosin, and the focal adhesion protein Talin (*Figures 6D and 7C, E, F*). These findings suggest that DAAM-depleted cells may adhere less well to the ECM. We previously showed that a mild reduction in cell–ECM adhesion increases the speed of follicle cell migration (*Lewellyn et al., 2013*). Similarly, mean migration rates for DAAM-depleted epithelia are faster than for control epithelia (*Figure 7G*). Altogether, these data suggest that DAAM contributes to the assembly of treadmilling SFs by adding new F-actin along their lengths, and/or by contributing F-actin to the basal cortex, and that this additional F-actin strengthens cell–ECM adhesion.

## Different formins build treadmilling versus canonical SFs

One striking feature of the SFs in the follicle cells is that they change in both structure and function between early developmental stages when they mediate collective cell migration and later stages when their contractility helps to create the elongated shape of the egg, so we next investigated the molecular mechanisms that underly this transition. We found that DAAM is downregulated after follicle cell migration has ceased (*Figure 8A*). This observation suggested that DAAM might mediate the formation of treadmilling SFs in migratory cells but not the more canonical SFs that form at later stages. To test this idea, we used traffic jam-Gal4 to express *DAAM RNAi* in the follicle cells throughout oogenesis. This reduces F-actin in the treadmilling SFs as expected. During postmigratory stages, however, F-actin levels in the SFs are unaffected (*Figure 8B*). Thus, DAAM selectively contributes to the formation of treadmilling SFs.

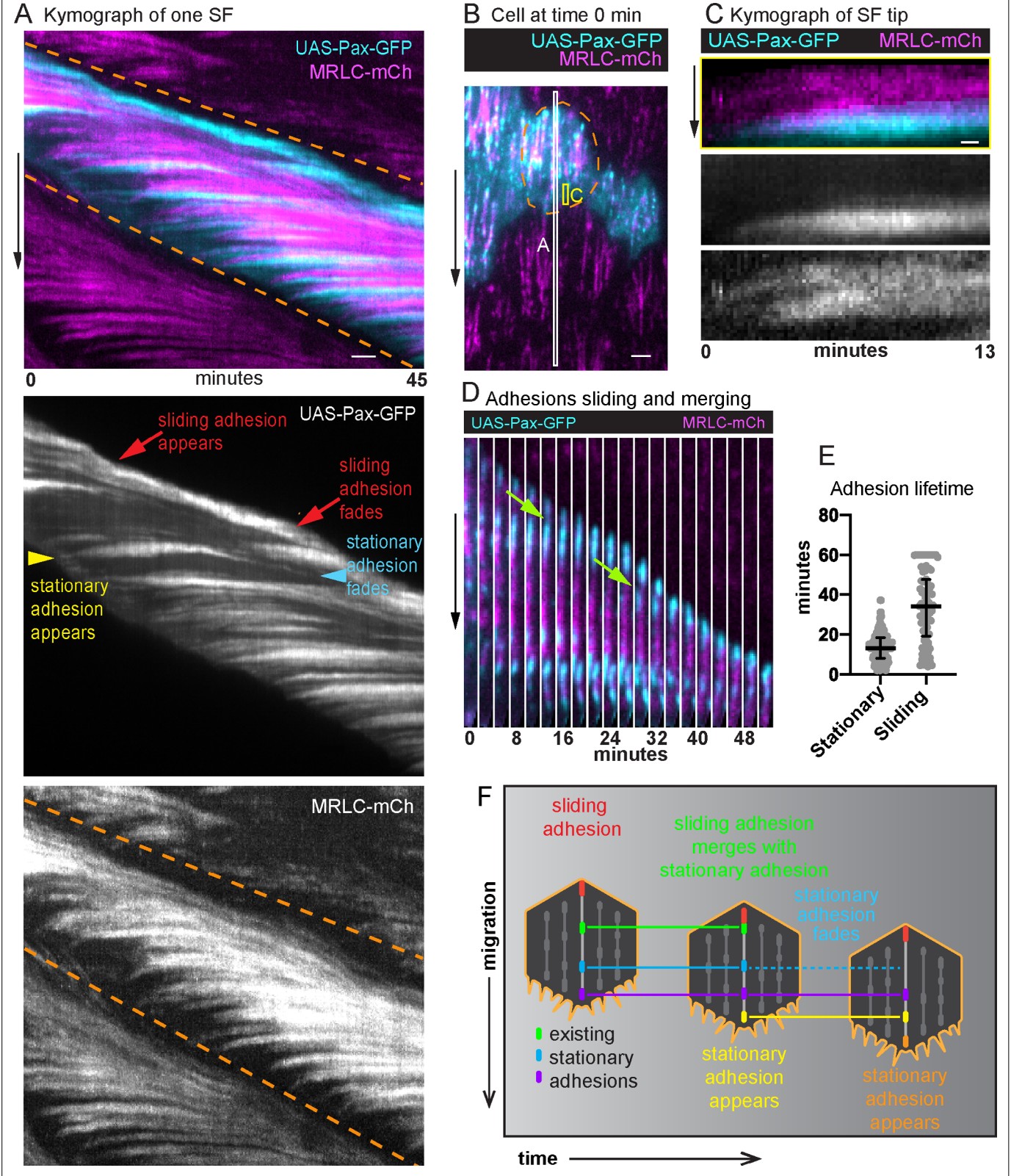

**Figure 4.** Adhesions are continuously added and removed from treadmilling stress fibers (SFs). (**A**) Kymograph of one SF from the white boxed region in (**B**) showing the appearance and disappearance of adhesions and actomyosin segments over time. The orange dashed lines mark the leading and trailing edges of the cell, respectively, and their diagonal orientation indicates the movement of the cell edges over time. The yellow arrowhead marks the addition of a new adhesion to the front of the SF. The slight rearward curve at the front of the trace suggests that the adhesion matures under

*Figure 4 continued on next page*

Figure 4 continued

tension. The trace then remains largely horizontal showing that the adhesion is stationary relative to the substrate. The trace fades as the cell rear approaches, showing adhesion disassembly. The many horizontal traces show that multiple adhesions are added to the same SF over time. Finally, the red arrows highlight the appearance and disappearance of a distinct adhesion at the rear. The diagonal orientation of this trace indicates that the final adhesion moves with the cell's trailing edge. The image shown is representative of kymographs for eight SFs. (B) Still image from a video of an epithelium in which all cells express myosin regulatory light chain (MRLC)-mCherry (mCh) and a subset of cells expresses UAS-Paxillin-GFP. Dashed line surrounds one cell. White and yellow boxes correspond to kymographs in (A) and (C), respectively. See *Figure 4—video 1*. (C) Kymograph of a SF tip from the yellow boxed region in (B), showing that Paxillin-GFP and MRLC-mCh levels increase in synchrony as the SF grows at the front. The image shown is representative of kymographs for seven SFs. (D) Still images from a video showing a sliding adhesion that persists for at least 50 min and merges with three stationary adhesions (green arrows). See *Figure 4—video 2*. (E) Quantification of adhesion lifetimes. In order on graph, n = 134, 84 adhesions from 23 cells in 3 egg chambers. Bars show medians and interquartile ranges. (F) Illustration summarizing adhesion dynamics in treadmilling SFs. Experiments performed at stage 7. Black arrows show migration direction. Scale bars: 2 µm (A, B), 0.5 µm (C).

The online version of this article includes the following video and figure supplement(s) for figure 4:

**Source data 1.** Excel file containing the numeric data used to generate the graph in *Figure 4E*.

**Figure 4—video 1.** Time-lapse video of a field of follicle cells.

https://elifesciences.org/articles/72881/figures#fig4video1

**Figure 4—video 2.** Time-lapse video of one stress fiber (SF) in which myosin is labeled with myosin regulatory light chain (MRLC)-mCherry (mCh) and adhesions labeled with UAS-Pax-GFP.

https://elifesciences.org/articles/72881/figures#fig4video2

Previous work suggested that the formin Dia is required for SF assembly in the follicle cells (*Delon and Brown, 2009*; *Popkova et al., 2020*). However, when we depleted Dia during migratory stages as part of our RNAi screen, the treadmilling SFs were unaffected (*Figure 8C* and *Figure 6—figure supplement 1D*). We know Dia was depleted because the RNAi-expressing clones contained multi-nucleated cells, consistent with Dia's role in cytokinesis. To ask if Dia selectively mediates canonical SF assembly during postmigratory stages, we expressed *Dia RNAi* using Cy2-Gal4, which initiates expression after cell divisions and cell migration have both ceased. This condition largely eliminates SFs from postmigratory cells, while similarly expressing *DAAM RNAi* does not (*Figure 8D, E*). These data show that there is a developmental switch in the formins that mediate SF assembly in the follicle cells, with DAAM contributing to treadmilling SFs during migratory stages and Dia contributing to the canonical SFs that form after migration is complete.

## Discussion

SFs are a common and well-studied feature of cells migrating on nonnative substrates in vitro. However, whether and how cells use SFs to migrate on their native substrates in vivo remains largely unexplored. Focusing on the follicular epithelial cells of *Drosophila*, we found that their SFs display a novel treadmilling behavior that allows individual SFs to persist as the cells migrate over more than one cell length. The discovery of these long-lived contractile structures has important implications for our understanding of SF dynamics, the influence that different SF types can have on cell motility, and how SF structure and function can change as a tissue develops, each of which is discussed below.

We found that SFs in migrating follicle cells have internal adhesions along their lengths, in addition to adhesions at the ends. These internal adhesions are key to the treadmilling behavior. When a new adhesion and actomyosin segment are added to the front of an existing SF, the previous front adhesion becomes an internal adhesion. In this way, the formation of internal adhesions depends on treadmilling. Once formed, the internal adhesions then contribute to treadmilling by generating the modular SF architecture needed for older stationary adhesions to be disassembled near the cell's rear and allow the back of the SF to shorten. To our knowledge this is the first study to focus on the role of internal adhesions in SF dynamics, but there are hints in the literature that these structures may exist in other cells. For example, SFs isolated from human foreskin fibroblasts and bovine endothelial cells have small puncta of the focal adhesion protein vinculin along their lengths in addition to prominent vinculin puncta at their ends (*Katoh et al., 1998*), and the cortical SFs in cultured mesenchymal cells were sometimes observed to have more than two adhesions (*Lehtimäki et al., 2021*). Internal adhesions have also been invoked as a possible explanation for the buckling pattern observed when SFs are rapidly compressed (*Costa et al., 2002*; *Kassianidou and Kumar, 2015*). Because internal

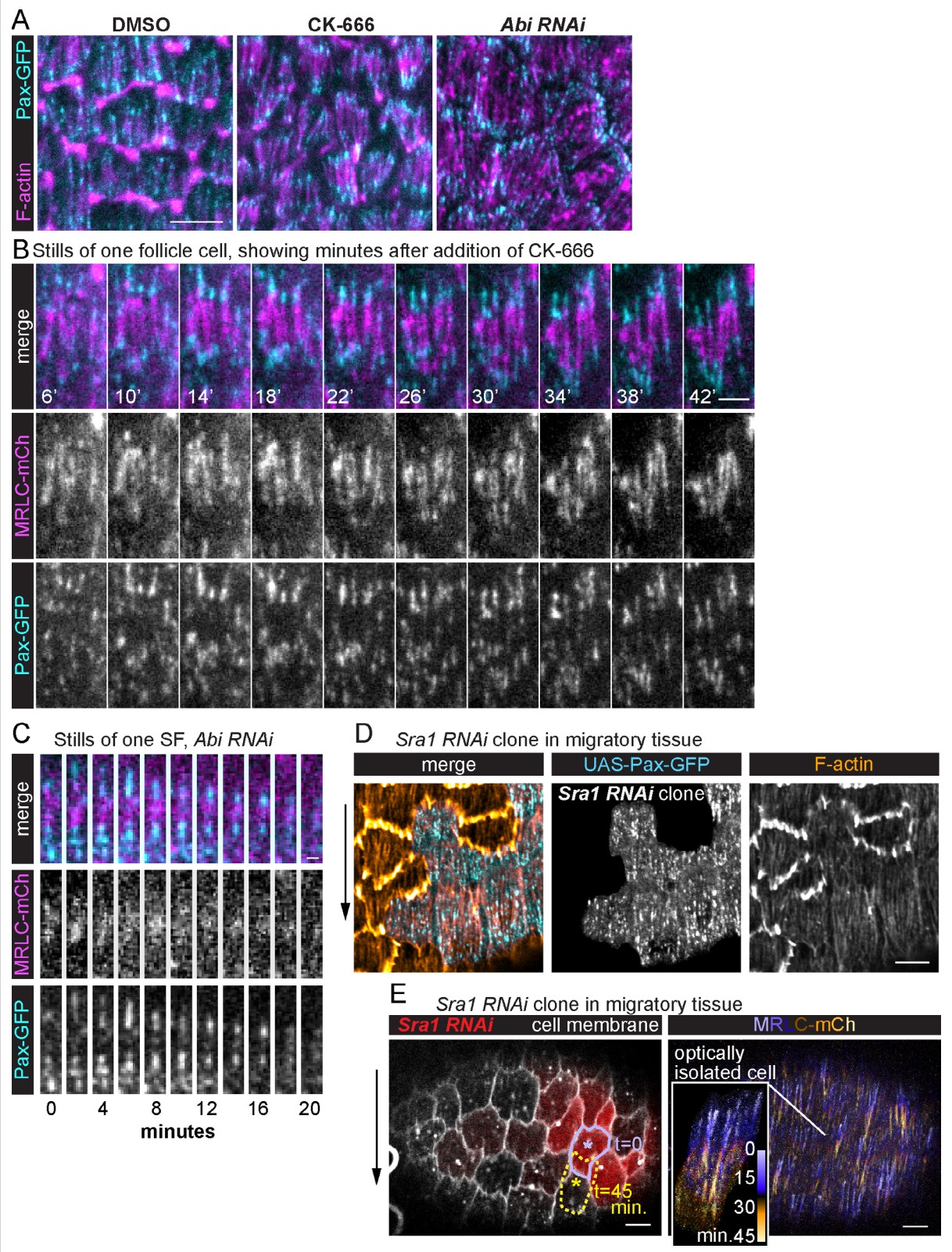

**Figure 5.** Stress fiber (SF) treadmilling depends on cell migration. (**A**) Images of epithelia in which migration has been blocked by eliminating leading-edge protrusions. Adhesions become concentrated at the SF ends. The images shown are representative of n = 28 (Dimethyl Sulfoxide [DMSO]), n = 29 (CK-666), and n = 16 (*Abi RNAi*) egg chambers that were analyzed. (**B**) Still images from a video of one cell showing that internal adhesions disappear, and end adhesions grow as addition of CK-666 slowly brings migration to a stop. The image shown is representative of eight egg chambers that were

*Figure 5 continued on next page*

*Figure 5 continued*

analyzed. See also *Figure 5—video 1*. (**C**) Still images from a video showing one SF in an epithelium in which migration has been blocked. The SF shortens and disappears with no new adhesions added to the ends. The images shown are representative of most SFs in eight egg chambers that were analyzed. (**D**) Image of a migrating epithelium with a clone of cells expressing *Sra1 RNAi* to eliminate protrusions. The SFs in the clone maintain internal adhesions. The image shown is representative of six egg chambers that were analyzed. (**E**) Still image from a video of a migrating epithelium with a clone of cells that expresses *Sra1 RNAi* to eliminate protrusions (left). Outline shows the movement of one cell over 45 min (lavender to yellow asterisks). Temporal projection of the SFs in the same epithelium at 20- s intervals (right). Inset shows SF growth in the *Sra1 RNAi* cell marked with the asterisk on the left. The image shown is representative of three egg chambers that analyzed. Experiments performed at stage 7. Black arrows show migration direction. Scale bars: 5 µm (**A, D, E**), 1 µm (**B**), 2 µm (**C**).

The online version of this article includes the following video for figure 5:

**Figure 5—video 1.** Time-lapse video of one follicle cell in which stress fibers (SFs) are labeled with myosin regulatory light chain (MRCL)-mCherry (mCh) and adhesions labeled with endogenously tagged Pax-GFP.

https://elifesciences.org/articles/72881/figures#fig5video1

adhesions are attached to two aligned actomyosin segments, they likely experience more balanced pulling forces than end adhesions, which may affect their composition and/or organization. It is also likely, however, that there is still higher tension on one side of the adhesion due to forces exerted by cell movement. Determining the extent to which internal adhesions are found in other cell types and how they differ from end adhesions represent important areas for future research.

A treadmilling SF grows at its front when a new adhesion and new actomyosin segment appear nearly simultaneously ahead of the foremost adhesion. How do these new elements arise? We envision that the new actomyosin segment captures a nearby nascent adhesion that has formed just behind the cell's leading edge, and that this activity both links the nascent adhesion to the existing SF and induces it to mature. Our finding that DAAM is required for robust SF formation also suggests two hypotheses for the source of the F-actin for the new actomyosin segment. One possibility is that DAAM localizes to nascent and mature adhesions, causing actin filaments to grow out from these sites, like the role ascribed to Dia in dorsal SF formation (*Hotulainen and Lappalainen, 2006*; *Oakes et al., 2012*; *Tojkander et al., 2011*). Bundling of the DAAM-generated filaments by myosin could then link the nascent adhesion to the mature adhesion. Alternatively, DAAM could play a more general role in contributing F-actin to the basal cortex along with other actin assembly factors (*Chugh and Paluch, 2018*), with local pulses of myosin activity near the cell's leading-edge condensing the cortical F-actin meshwork to form the new actomyosin segment. We favor the second model for two reasons. First, DAAM is found throughout the cortex with no obvious enrichment at adhesions. Second, treadmilling SFs closely resemble other SF types that arise from the cortex and/or are embedded within it (*Lehtimäki et al., 2021*; *Svitkina, 2020*; *Vignaud et al., 2020*), as they lie so flat against the basal surface that we can visualize them with near-TIRF microscopy. This contrasts with ventral SFs whose center can arch away from the migratory surface. However, future work will be required to distinguish between these possibilities.

A treadmilling SF shrinks at its back using a mechanism involving an adhesion that slides along the substrate as the actomyosin segment in front of it shortens. These rear-most sliding adhesions are strikingly similar to those found in individual migrating cells in culture (*Ballestrem et al., 2001*; *Digman et al., 2008*; *Laukaitis et al., 2001*; *Rid et al., 2005*; *Smilenov et al., 1999*; *Wehrle-Haller and Imhof, 2003*), in that they primarily form near the cell's trailing edge and then track with its movement. In migrating cells, sliding adhesions have been proposed to act as rudders that help to steer the cell (*Rid et al., 2005*). In stationary cells, sliding adhesions can also remodel the ECM (*Lu et al., 2020*; *Zamir et al., 2000*). Given that one of the purposes of follicle cell migration is to polarize the basement membrane ECM over which they move (*Gutzeit, 1991*; *Haigo and Bilder, 2011*; *Isabella and Horne-Badovinac, 2016*), it is interesting to speculate that sliding adhesions could function in both roles in these cells.

The modular architecture and unidirectional growth of a treadmilling SF both depend on cell movement. We found this relationship by comparing two conditions that eliminate leading-edge protrusions. In one condition, we eliminated protrusions from a clone of cells. Here, the epithelium continues to migrate, carrying the nonprotrusive cells along for the ride (*Cetera et al., 2014*). The SFs in the clone are indistinguishable from those in wild-type cells, showing that protrusive F-actin networks are not required for treadmilling SFs to form. In the other condition, we eliminated protrusions from the

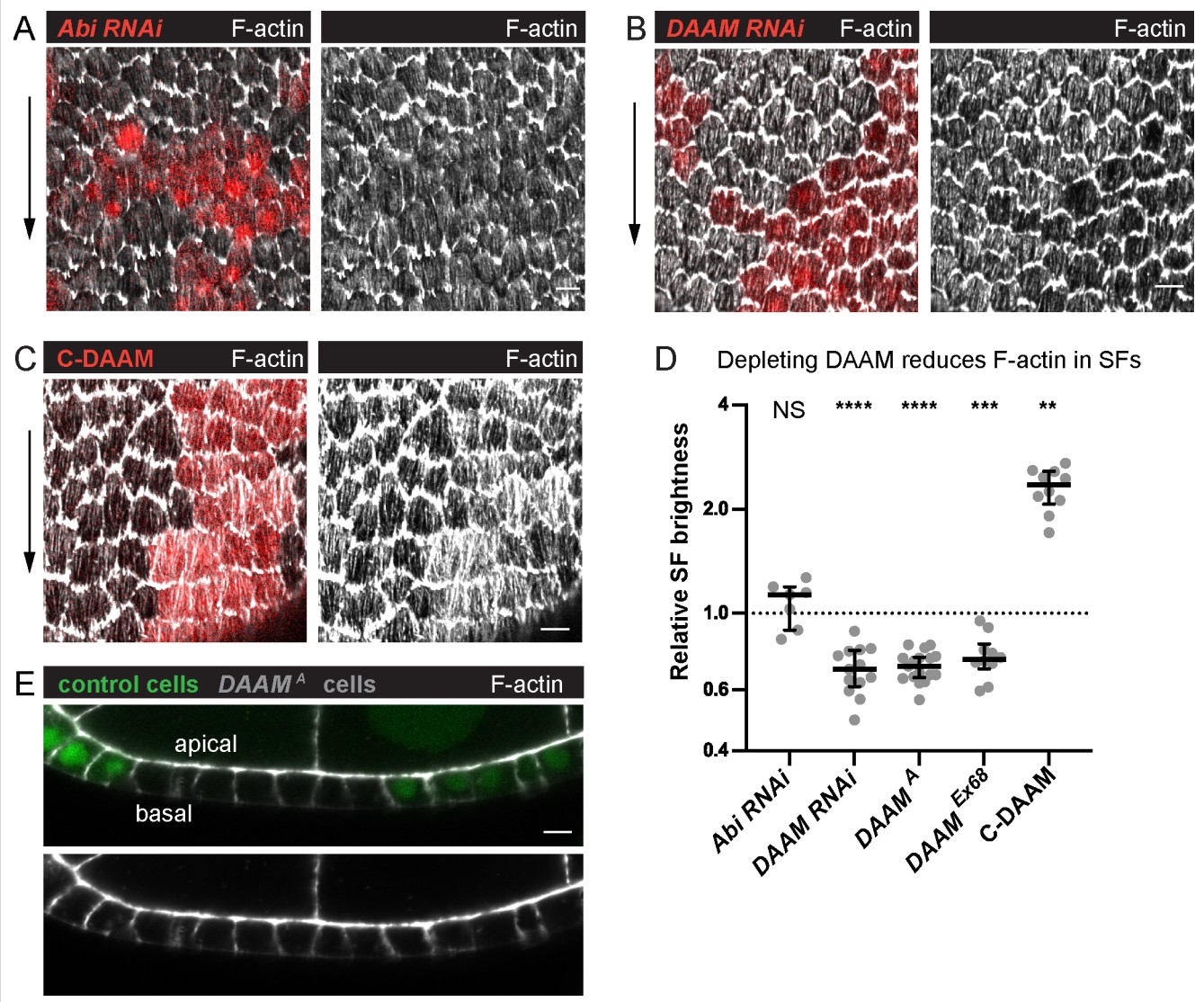

**Figure 6.** Dishevelled-associated activator of morphogenesis (DAAM) contributes to treadmilling stress fiber (SF) assembly. (**A–C**) Images of epithelia with clones of cells expressing various transgenes. (**A**) *Abi RNA*i reduces F-actin in protrusions but not SFs. (**B**) *DAAM RNA*i reduces F-actin in SFs. (**C**) An activated form of DAAM (C-DAAM) increases F-actin in SFs. (**D**) Quantification of the data in (**A–C**). Each point is the ratio of the mean value for F-actin levels in SFs from 10 experimental cells and 10 nearby control cells in the same egg chamber. In order on graph, *n* = 7, 13, 17, 10, 10 egg chambers. Bars show medians and interquartile ranges. Two-tailed Wilcoxon matched pairs signed ranks test. NS (not significant) p > 0.05, \*\*p < 0.01, \*\*\*p < 0.001, \*\*\*\*p < 0.0001. See *Figure 6—figure supplement 1* for tests of additional actin assembly factors. (**E**) Image of a transverse section through an epithelium with a clone of *DAAM*^A mutant cells, representative of 11 egg chambers. Loss of DAAM does not obviously reduce cortical F-actin on lateral or apical surfaces. Experiments performed at stage 7. Black arrows show migration direction. Scale bars: 5 µm.

The online version of this article includes the following figure supplement(s) for figure 6:

**Source data 1.** Excel file containing the numeric data and exact p values used to generate the graphs in *Figure 6* and *Figure 6—figure supplement 1*.

**Figure supplement 1.** A screen for F-actin assembly factors that mediate treadmilling stress fiber (SF) formation.

entire tissue, which does block epithelial migration. Here, the SFs transition to having large adhesions only at their ends, but they are not stationary. Instead, when a new SF forms, it continuously shortens toward the middle until it disappears with no new material added to the ends. This movement is reminiscent of the way a treadmilling SF normally shortens at its rear, except that it happens from both ends. This raises the possibility that these SFs have lost their polarity and have two 'backs'. If true, studies of these aberrant SF dynamics could help to reveal how a treadmilling SF becomes polarized for unidirectional movement.

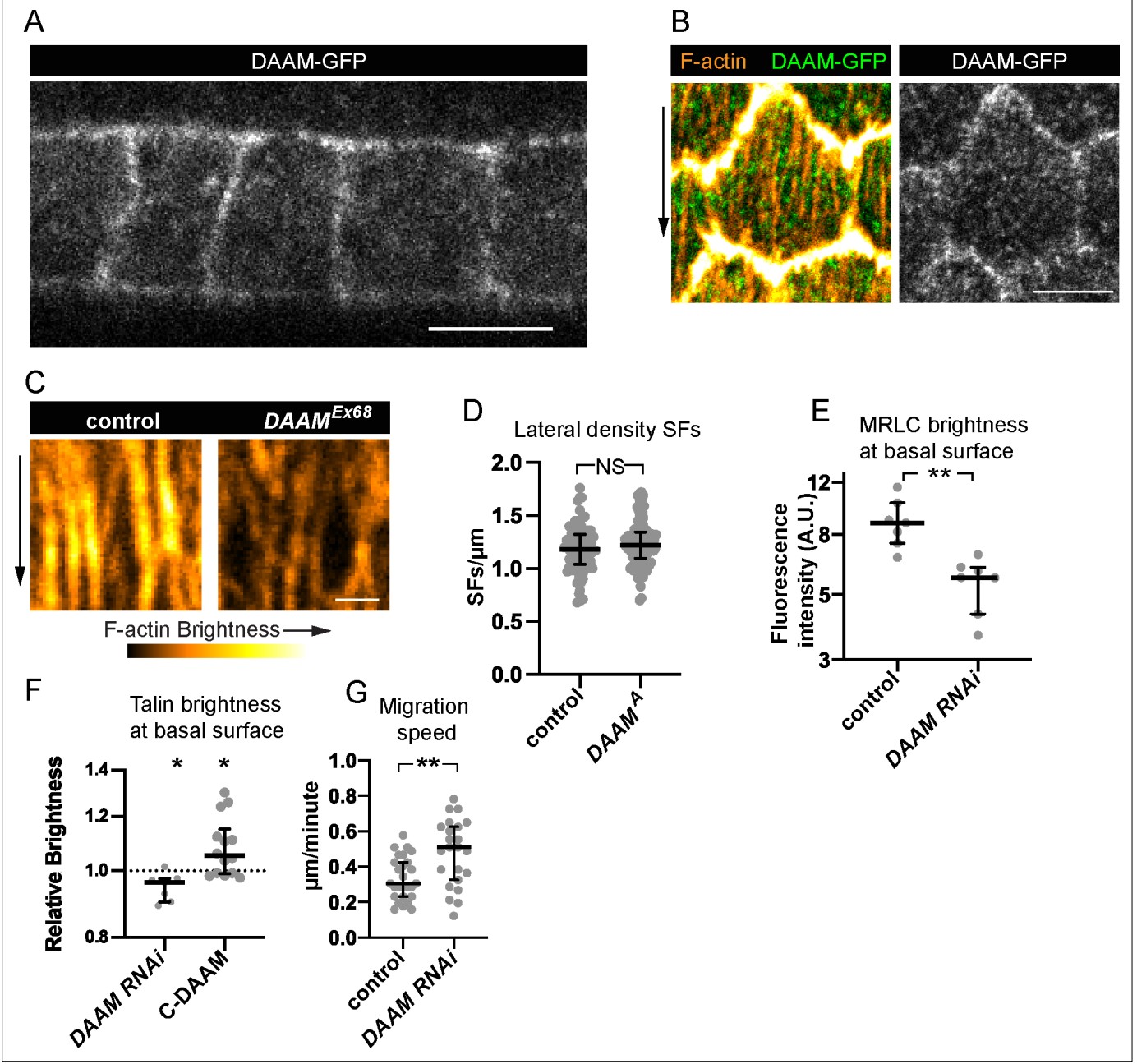

**Figure 7.** Dishevelled-associated activator of morphogenesis (DAAM) localizes to the cortex and likely strengthens cell–extracellular matrix (ECM) adhesion via stress fibers (SFs). (**A, B**) Images of DAAM-GFP (endogenous tag). (**A**) Transverse section showing that DAAM localizes to the entire cell cortex, image representative of five egg chambers. (**B**) Basal view of one cell showing DAAM relative to SFs, image representative of five egg chambers. (**C**) Images of SFs from a control cell and *DAAM^{Ex68}* cell in the same epithelium stained with phalloidin. SFs in *DAAM^{Ex68}* cells are similar in number but have reduced F-actin fluorescence. Image representative of three egg chambers. (**D**) Quantification showing that lateral SF density is normal in *DAAM^A* cells. In order on graph, *n* = 90, 90 cells from 9 egg chambers with mitotic clones. (**E**) Quantification showing that myosin regulatory light chain (MRLC) levels are reduced in *DAAM RNAi* cells. In order on graph, *n* = 7, 7 egg chambers. (**F**) Quantification showing that Talin levels are reduced in *DAAM RNAi* cells. Each point is the ratio of the mean value for Talin levels from 10 experimental cells and 10 control cells in the same egg chamber. In order on graph, *n* = 7, 14 egg chambers. (**G**) Quantification of migration rates for control and *DAAM RNAi* epithelia. In order on graph, *n* = 27, 23 egg chambers. Experiments performed at stage 7. Black arrows show migration direction. Scale bars: 5 µm (**A, B**), 1 µm (**C**). (**E–H**) Bars show medians and interquartile ranges. NS (not significant) p > 0.05, *p < 0.05, **p < 0.01. (**D, E, G**) Two-tailed Mann–Whitney test. (**F**) Two-tailed Wilcoxon matched pairs signed ranks test.

The online version of this article includes the following source data for figure 7:

**Source data 1.** Excel file containing the numeric data and exact p values used to generate the graphs in *Figure 7*.

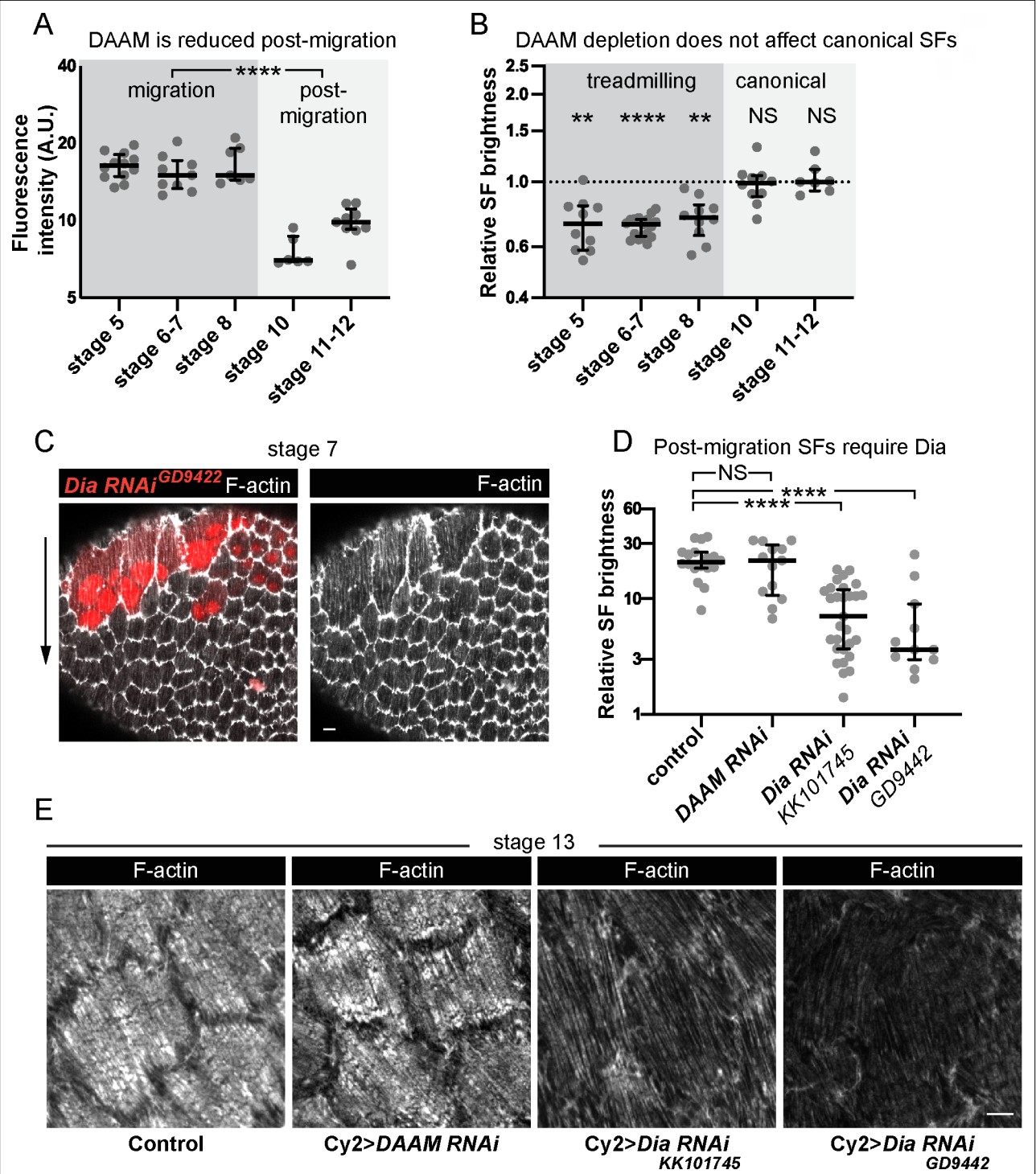

**Figure 8.** Different formins contribute to treadmilling versus canonical stress fibers (SFs). (**A**) Quantification of Dishevelled-associated activator of morphogenesis (DAAM)-GFP levels at basal surface. Levels are higher during migration stages. In order on graph, *n* = 12, 9, 7, 6, 9 egg chambers. Statistics are on pooled data from migration versus postmigration stages. Two-tailed Mann–Whitney test. (**B**) Quantification of F-actin levels in SFs. Loss of DAAM reduces F-actin in treadmilling SFs, but not canonical SFs. Each point is the ratio of the mean value for 10 RNAi cells and 10 control cells in the same egg chamber. In order on graph, *n* = 10, 17, 10, 10, 7 egg chambers. Statistics by stage are two-tailed Wilcoxon matched pairs signed ranks tests. (**C**) Image of an epithelium with a clone of cells expressing *Dia RNAi*. *Dia RNAi* cells have cytokinesis defects shown by multiple red nuclei per cell, but F-actin levels in SFs are normal. (**D**) Quantification of F-actin levels in SFs at stage 13. When Cy2-Gal4 is used to drive RNAi during postmigration stages, *Dia RNAi* reduces F-actin in SFs, but *DAAM RNAi* does not. In order on graph, *n* = 18, 13, 29, 11 egg chambers. Two-tailed Mann–Whitney tests. (**E**) Images of cells showing that *Dia RNAi* reduces F-actin levels in SFs postmigration while *DAAM RNAi* does not (images selected from intermediate

*Figure 8 continued on next page*

*Figure 8 continued*

brightness values measured in D). Scale bars: 5 µm. (**A, B, D**) Bars show medians and interquartile ranges. NS (not significant) p > 0.05, **p < 0.01, ****p < 0.0001.

The online version of this article includes the following figure supplement(s) for figure 8:

**Source data 1.** Excel file containing the numeric data and exact p values used to generate the graphs in *Figure 8*.

We propose that treadmilling SFs may be particularly well suited to mediate the collective migration of epithelial cells. Cells that migrate as individuals can undergo frequent turns as they explore their environment. By contrast, each follicle cell follows a roughly linear trajectory over its entire migratory period, which can last for up to 2 days of egg chamber development (*Cetera et al., 2014*; *Horne-Badovinac and Bilder, 2005*). The follicle cells use intercellular signaling to align all their front–rear axes in the same direction across the tissue (*Barlan et al., 2017*; *Stedden et al., 2019*). Once this alignment is achieved, however, the long lifetimes and unidirectional growth of their SFs likely reinforce this tissue-scale order by ensuring that each cell maintains the linear trajectory required for the entire epithelium to move in a directed way. Given that the collective migration of epithelial cells plays central roles in morphogenesis, turnover of the intestinal lining, wound repair, and the metastatic cascade (*Friedl and Gilmour, 2009*; *Jain et al., 2021*; *Mishra et al., 2019*; *Scarpa and Mayor, 2016*), the use of treadmilling SFs to direct cell motility in natural contexts may be widespread.

Finally, this work highlights how SFs within a given cell type can change in both their structure and mode of assembly as a tissue develops. When follicle cell migration ends, the SFs take on a new morphogenetic role in which their contractile activity helps to create the elongated shape of the egg (*Cerqueira Campos et al., 2020*; *He et al., 2010*). This change in function is accompanied by a change in SF organization, in which the density of SFs across the basal surface increases, internal adhesions disappear, and large focal adhesions become concentrated at the fibers' ends. Previous work revealed that the focal adhesions associated with the two SF types differ, with αPS1/βPS being present during migratory stages and αPS2/βPS taking over after migration is complete; tensin is also only present in postmigratory stages (*Delon and Brown, 2009*). We have now found that even the mode of F-actin assembly differs, with DAAM mediating the formation of SFs in migratory cells and Dia mediating their formation postmigration, a result that is consistent with previous studies of Dia in the follicle cells (*Delon and Brown, 2009*; *Popkova et al., 2020*). Why a different formin provides the F-actin for each type of SF is not immediately clear. However, this observation further underscores the importance of studying SFs in developing tissues where such structural transitions can and do occur.

Altogether, this work defines a new type of long-lived, treadmilling SF that appears to be ideally suited for collectively migrating epithelial cells in vivo. It further highlights how studying SFs in a natural context can reveal unexpected changes in the mechanisms that underly their assembly as a tissue develops.

## Materials and methods

**Key resources table**

| Reagent type (species) or resource | Designation | Source or reference | Identifiers | Additional information |
|---|---|---|---|---|
| Antibody | anti-GFP directly coupled to Alexa Fluor 488 (rabbit polyclonal) | Invitrogen | Cat # A21311 | (1:400) |
| Chemical compound, drug | CellMask Deep Red Plasma Membrane Stain | Thermo Fisher Scientific | Cat# C10046 | (1:1000) |
| Chemical compound, drug | Alexa Fluor 647 phalloidin | Thermo Fisher Scientific | Cat# A22287 | (1:100) |
| Chemical compound, drug | Alexa Fluor 488 phalloidin | Thermo Fisher Scientific | Cat# A12379 | (1:200) |
| Chemical compound, drug | Schneider's *Drosophila* medium | Thermo Fisher Scientific | Cat# 21720-024 | |

*Continued on next page*

*Continued*

| Reagent type (species) or resource | Designation | Source or reference | Identifiers | Additional information |
|---|---|---|---|---|
| Chemical compound, drug | Fetal bovine serum | Gibco | Cat# 10438-018 | |
| Chemical compound, drug | Recombinant Human Insulin | Millipore Sigma | Cat# 12,643 | |
| Other | Soda Lime Glass Beads, 48–51 µm | Cospheric LLC | Cat# S-SLGMS-2.5 | |
| Chemical compound, drug | Formaldehyde, 16%, methanol free, Ultra-Pure | Polysciences | Cat# 18814-10 | |
| Chemical compound, drug | SlowFade Diamond Antifade mounting medium | Invitrogen | Cat# S36972 | |
| Chemical compound, drug | SlowFade Antifade Kit | Thermo Fisher Scientific | Cat# S2828 | |
| Chemical compound, drug | CK-666 | Millipore Sigma | Cat# SML0006 | |
| Software, algorithm | ImageJ version 2.1.0/1.53c | | https://fiji.sc/ | |
| Software, algorithm | Handbrake 1.3.3 The open source video transcoder | HandBrake Team | https://handbrake.fr/ | |
| Software, algorithm | Zen Blue | Zeiss | | |
| Software, algorithm | Zen Black | Zeiss | | |
| Software, algorithm | MetaMorph | Molecular Devices | | |
| Software, algorithm | Prism Version 8 | Graphpad | | |
| Genetic reagent (*D. melanogaster*) | w[1118] | Bloomington *Drosophila* Stock Center | BDSC: 3605; FlyBase ID: FBst0003605; RRID: BDSC_3605 | FlyBase symbol: w[1,118] |
| Genetic reagent (*D. melanogaster*) | sqh-GFP | Vienna *Drosophila* Resource Center | VDRC: 318484; FlyBaseID: FBst0492100 | FlyBase symbol: PBac{fTRG00600.sfGFP-TVPTBF}VK00033 |
| Genetic reagent (*D. melanogaster*) | MRLC-mCh | Laboratory of Eric Wieschaus (*Martin et al., 2009*) | | sqhAx3/FM7;; sqh> sqh-mCh/TM3, Ser, actGFP |
| Genetic reagent (*D. melanogaster*) | UAS-Pax-GFP | Laboratory of Denise Montell (*He et al., 2010*) | | |
| Genetic reagent (*D. melanogaster*) | daughterless-Gal4 | Bloomington *Drosophila* Stock Center | BDSC: 55851; FlyBaseID: FBst0055851; RRID:BDSC_55851 | FlyBase symbol: w*; P{GAL4-da.G32}UH1, Sb1/TM6B, Tb1 |
| Genetic reagent (*D. melanogaster*) | hs-FLP | Bloomington *Drosophila* Stock Center | BDSC: 8862; FlyBase ID: FBst0008862; RRID:BDSC_8862 | FlyBase symbol: P{ry[+ t7.2] = hsFLP}22, w[*]} |
| Genetic reagent (*D. melanogaster*) | act5c >> Gal4 | Bloomington *Drosophila* Stock Center | BDSC: 4780; FlyBase ID: FBst0004780; RRID:BDSC_4780 | FlyBase symbol: y1 w*; P{GAL4-Act5C(FRT. CD2).P}S |
| Genetic reagent (*D. melanogaster*) | UAS-Ftractin-Tom | Bloomington *Drosophila* Stock Center | BDSC: 58989; FlyBaseID: FBtp0095457; RRID:BDSC_4780 | P{UASp-F-Tractin.tdTomato}15 A/SM6b; MKRS/TM2 |
| Genetic reagent (*D. melanogaster*) | UAS-Lifeact-GFP | Bloomington *Drosophila* Stock Center | BDSC: 35544; FlyBaseID: FBst0035544; RRID:BDSC_35544 | FlyBase symbol: y1 w*; P{UAS-Lifeact-GFP} VIE-260B |

*Continued on next page*

*Continued*

| Reagent type (species) or resource | Designation | Source or reference | Identifiers | Additional information |
|---|---|---|---|---|
| Genetic reagent (*D. melanogaster*) | UASMoesinABD-mCh 42c | Laboratory of Brooke McCartney | | |
| Genetic reagent (*D. melanogaster*) | UAS-Utrophin-ABD-GFP | Laboratory of Thomas Lecuit (*Rauzi et al., 2010*) | | |
| Genetic reagent (*D. melanogaster*) | Pax-GFP | Kyoto Stock Center | DGRC: 109971; FlyBaseID: FBst0325098 | FlyBase symbol: w[1,118]; PBac{EGFP-IV} Pax[KM0601] |
| Genetic reagent (*D. melanogaster*) | Talin-GFP | Laboratory of Hugo Bellen (*Venken et al., 2011*) | BDSC: 39649; FlyBaseID: FBst0039649; RRID:BDSC_39649 | FlyBase symbol: y1 w*; Mi{PT-GFSTF.0}rheaMI00296-GFSTF.0 lncRNA:CR43910MI00296-GFSTF.0-X |
| Genetic reagent (*D. melanogaster*) | ß-PS integrin-GFP | Laboratory of Nicholas Brown (*Klapholz et al., 2015*) | | |
| Genetic reagent (*D. melanogaster*) | traffic jam-Gal4 | Kyoto Stock Center | DGRC: 104055; FlyBaseID: FBst0302922 | FlyBase symbol: y* w*; P{w + mW.hs=GawB} NP1624/ CyO, P{w-=UASlacZ. UW14}UW14 |
| Genetic reagent (*D. melanogaster*) | UAS-Abi RNAi | National Institute of Genetics, Japan | NIG: 9749 R-3 | |
| Genetic reagent (*D. melanogaster*) | UAS-Sra1 RNAi | Bloomington *Drosophila* Stock Center | BDSC: 38294; FlyBaseID: FBst0038294; RRID:BDSC_38294 | FlyBase symbol: y1 sc* v1 sev21; P{TRiP. HMS01754}attP2 |
| Genetic reagent (*D. melanogaster*) | act5c >> Gal4, UAS-RFP | Bloomington *Drosophila* Stock Center | BDSC: 30558; FlyBase ID: FBst0030558; RRID:BDSC_30558 | FlyBase symbol: w[1,118]; P{w[+mC] = GAL4-Act5C(FRT.CD2).P}S, P{w[+mC] = UAS RFP.W}3/TM3, Sb (*Ballestrem et al., 2001*) |
| Genetic reagent (*D. melanogaster*) | UAS-pTWFlag-C-DAAM | Laboratory of József. Mihály (*Matusek et al., 2006*) | | |
| Genetic reagent (*D. melanogaster*) | hsFLP RFP FRT 19A | Bloomington *Drosophila* Stock Center | BDSC: 31418; FlyBaseID: FBst0031418; RRID: BDSC_31418 | FlyBase symbol: Ubi-mRFP.nls, w*, hsFLP neoFRT19A |
| Genetic reagent (*D. melanogaster*) | DAAM[A] FRT 19A | Bloomington *Drosophila* Stock Center | BDSC: 52348; FlyBaseID: FBst0052348; RRID: BDSC_52348 | FlyBase symbol: y1 DAAMA w* P{neoFRT}19 A/FM7c, P{GAL4-Kr.C}DC1, P{UAS-GFP.S65T}DC5, sn+ |
| Genetic reagent (*D. melanogaster*) | DAAM[Ex68] | Laboratory of József. Mihály (*Dollar et al., 2016*) | | |
| Genetic reagent (*D. melanogaster*) | 19A FRT | Bloomington *Drosophila* Stock Center | BDSC: 1709; FlyBaseID: FBst0001709; RRID: BDSC_1709 | FlyBase symbol: P{ry[+ t7.2] = neoFRT}19A; ry[506] |
| Genetic reagent (*D. melanogaster*) | DAAM[Ex68] FRT 19A | Recombination only, this study | DAAM[Ex68] from J. Mihály; and 19A FRT from BDSC: 1,709 | |
| Genetic reagent (*D. melanogaster*) | RFP FRT 19A | Bloomington *Drosophila* Stock Center | BDSC: 31416; FlyBase ID: FBst0031416; RRID: BDSC_31416 | FlyBase symbol: P{w[+ mC] = Ubi mRFP.nls}1, w[1,118], P{ry[+ t7.2] = neoFRT}19A |

*Continued on next page*

*Continued*

| Reagent type (species) or resource | Designation | Source or reference | Identifiers | Additional information |
|---|---|---|---|---|
| Genetic reagent (*D. melanogaster*) | UAS-DAAM RNAi | Vienna *Drosophila* Resource Center | VDRC: 103921; FlyBase ID: FBst0475779 | FlyBase symbol: P{KK102786}VIE-260B |
| Genetic reagent (*D. melanogaster*) | UAS-Dia RNAi | Vienna *Drosophila* Resource Center | VDRC: 103914; FlyBase ID: FBst0475772 | FlyBase symbol: P{KK101745}VIE-260B |
| Genetic reagent (*D. melanogaster*) | UAS-Dia RNAi | Vienna *Drosophila* Resource Center | VDRC: 20518; FlyBase ID: FBst0453738 | FlyBase symbol: w1118; P{GD9442}v20518 |
| Genetic reagent (*D. melanogaster*) | UAS-FHOS RNAi | Vienna *Drosophila* Resource Center | VDRC: 45,838 (line has been discontinued) | w[1,118]; +; P{GD10435}v145838 |
| Genetic reagent (*D. melanogaster*) | UAS-FHOS RNAi | Vienna *Drosophila* Resource Center | VDRC: 34034; FlyBase ID: FBst0460422 | FlyBase symbol: w1118; P{GD10435}v34034 |
| Genetic reagent (*D. melanogaster*) | UAS-Frl RNAi | Vienna *Drosophila* Resource Center | VDRC: 34413; FlyBase ID: FBst0460614 | FlyBase symbol: w1118; P{GD10799}v34413 |
| Genetic reagent (*D. melanogaster*) | UAS-Frl RNAi | Vienna *Drosophila* Resource Center | VDRC: 110438; FlyBase ID: FBst0482010 | FlyBase symbol: P{KK101703}VIE-260B |
| Genetic reagent (*D. melanogaster*) | UAS-Capu RNAi | Vienna *Drosophila* Resource Center | VDRC: 110404; FlyBase ID: FBst0481976 | FlyBase symbol: P{KK101400}VIE-260B |
| Genetic reagent (*D. melanogaster*) | UAS-Capu RNAi | Vienna *Drosophila* Resource Center | VDRC: 34278; FlyBase ID: FBst0460552 | FlyBase symbol: w1118; P{GD879}v34278 |
| Genetic reagent (*D. melanogaster*) | UAS-Form3 RNAi | Vienna *Drosophila* Resource Center | VDRC: 45594; FlyBase ID: FBst0466234 | FlyBase symbol: w1118; P{GD12856}v45594 |
| Genetic reagent (*D. melanogaster*) | UAS-Form3 RNAi | Vienna *Drosophila* Resource Center | VDRC: 107473; FlyBase ID: FBst0479293 | FlyBase symbol: P{KK110697}VIE-260B |
| Genetic reagent (*D. melanogaster*) | UAS-Ena RNAi | Vienna *Drosophila* Resource Center | VDRC: 43058; FlyBase ID: FBst0464896 | FlyBase symbol: w1118; P{GD8910}v43058/CyO |
| Genetic reagent (*D. melanogaster*) | w[1118] DAAM-GFP | Laboratory of József. Mihály (*Molnár et al., 2014*) | | |
| Genetic reagent (*D. melanogaster*) | UAS-Dcr | Bloomington *Drosophila* Stock Center | BDSC: 24651; FlyBase ID: FBst0024651; RRID:BDSC_24651 | FlyBase symbol: w1118; P{UAS-Dcr-2.D}10 |
| Genetic reagent (*D. melanogaster*) | Cy2 Gal4 | Laboratory of Nir Yakoby (*Queenan et al., 1997*) | FlyBase ID: FBti0007266 | FlyBase symbol: Dmel\P{GawB}CY2 |

## *Drosophila* genetics

We cultured *D. melanogaster* on cornmeal molasses agar food using standard techniques and performed all experiments on adult females. For most experiments, we raised crosses at 25°C and aged experimental females on vials sprinkled with dry yeast with males for 2–3 days at the same temperature or at 29°C, as noted in *Supplementary file 1*. For tissue-wide depletion of Abi, which causes round eggs that block the ovary, we dissected females no more than 2 days after eclosion. Experimental genotypes for each figure panel are in *Supplementary file 1*.

We produced clones of either *DAAM^A^* or *DAAM^Ex68^* mutant cells using FRT19A with the heat shock promoter driving FLP recombinase expression. For Flp-out clones, we crossed UAS lines to flies with FLP recombinase under a heat shock promoter and an Act5c >> Gal4 Flp-out cassette with or without UAS-RFP. We induced heat shock by incubating pupae and adults at 37°C for 1 hr, followed by 1 hr of recovery at 25°C, and then another hour at 37°C. We performed this heat shock procedure approximately six times over the course of 3–4 days.

Stocks are from the Bloomington *Drosophila* Stock Center or the Vienna *Drosophila* Resource Center (see *Supplementary file 1* for details) with the following exceptions: traffic jam-Gal4 (104-055),

UAS-*Abi RNAi* (NIG9749R-3), and Pax-GFP (109-971) are from the *Drosophila* Genetic Resource Center in Kyoto; UAS-Utr-ABD-GFP is a gift of Thomas Lecuit (*Rauzi et al., 2010*), UAS-C-DAAM (*Matusek et al., 2006*), DAAM^Ex68 (*Dollar et al., 2016*), and DAAM-GFP (*Molnár et al., 2014*) are gifts from József Mihály; UAS-Moe-ABD-mCh is a gift of Brooke McCartney; UAS-Pax-GFP is a gift of Denise Montell (*He et al., 2010*), MRLC-mCh is a gift of Eric Wieschaus (*Martin et al., 2009*), βPS integrin/Mys-GFP is a gift from Nick Brown (*Klapholz et al., 2015*), Talin-GFP is gift from Hugo Bellen (*Venken et al., 2011*), and Cy2-Gal4 is a gift from Nir Yakoby (*Queenan et al., 1997*).

## Time-lapse video acquisition and microscopy

We performed live imaging of egg chambers largely as described (*Cetera et al., 2016*), with the exact procedure outlined below. We collected experimental females 0–2 days after eclosion and aged them on yeasted fly food for 1–2 days. We dissected ovaries in live imaging media (Schneider's *Drosophila* medium containing 15% fetal bovine serum and 200 mg/ml recombinant human insulin [Sigma]), also in some experiments containing CellMask Green (Thermo Fisher; 1:500), or Orange or Deep Red Plasma Membrane Stain (Thermo Fisher; 1:1000). After carefully removing the muscle sheaths with forceps, we transferred individual ovarioles to fresh live imaging media to wash out excess CellMask, then transferred 30 µl of ovarioles and media to a glass slide, adding 51 µm Soda Lime Glass beads (Cospheric LLC) to support a 22 × 30 mm No. 1.5 coverslip. We sealed the edges of the coverslip with Vaseline to prevent evaporation. Each slide was used for no more than 1.5 hr. We examined all egg chambers for damage prior to imaging.

We imaged egg chambers using a Nikon ECLIPSE-Ti inverted microscope equipped with a Ti-ND6-PFS Perfect Focus Unit. A laser merge module (Spectral Applied Research) controlled 481 and 561 nm laser excitation from 50 mW solid-state lasers (Coherent Technology) to a motorized TIRF illuminator. We adjusted the laser illumination angle to achieve near-TIRF illumination (*Tokunaga et al., 2008*). We collected images using a Nikon CFI ×100 Apo 1.45 NA oil immersion TIRF objective combined with ×1.5 intermediate magnification onto an Andor iXon3 897 EMCCD camera. All image acquisition was controlled using MetaMorph software. We obtained time-lapse videos by capturing single planes near the basal epithelial surface every 10–30 s.

We used ImageJ (*Schindelin et al., 2012*; *Schneider et al., 2012*) to set minimum and maximum pixel values, and in *Figure 1A* to perform gamma adjustment on the original 16-bit image data, before converting images to 8-bit grayscale format for display. We performed these operations identically for all images that are compared directly.

## Analyses from live imaging data

To calculate epithelial migration rates, we generated kymographs from the time-lapse image stacks in ImageJ (*Schindelin et al., 2012*; *Schneider et al., 2012*) by drawing a single line across several cell diameters in the direction of migration. We determined the migration rate for each epithelium by measuring the slope of 3–4 kymograph lines and averaging the values. Please see *Barlan et al., 2017* for an illustration of this technique.

To measure lifetimes of individual SFs, we began at the midpoint of hour-long videos (imaged at 10- or 20- s intervals) of egg chambers expressing MRLC-mCh or -GFP. We selected one SF at a time from cells that remained in view for the entire video and ran the video backwards to identify its frame of origin (if any) and forwards to identify its frame of disappearance (if any). Because egg chambers kept on slides for longer than an hour often exhibited a slowing of migration, we did not attempt to measure SF lifetime from longer videos. We expressed SF lifetime both in minutes and in cell lengths by calculating each egg chamber's migration speed as described above, and by measuring front–rear cell length at the cell's middle at the video's midpoint.

To measure lifetimes and behavior of individual adhesions, we imaged egg chambers expressing MRLC-mCh and Pax-GFP under the patchy driver da-Gal4 (to allow delineation of individual cells) for 1 hr at 10- or 20- s intervals. As described for SFs above, we identified adhesions in the midpoint of these videos and followed them frame-by-frame, backwards and forwards, to find their time of origin and disappearance. Two mostly distinct populations of adhesions were found, one that remained in place after forming near the front or in the middle of cells ('stationary adhesions') and one that slid at the same rate as the migrating cells, invariably found near the rear ('sliding adhesions'). Their lifetime data were tabulated separately. Sliding adhesions frequently exhibited merging behavior with

stationary adhesions, and this, as well as the proportion of SFs that had a sliding adhesion at a given time, was noted.

To produce the kymographs shown in *Figure 4*, we aligned images so that the axis of the SF coincided with the vertical (*y*) image axis. For *Figure 4A*, we selected rectangular regions whose width (in *x*) coincided with the width of a single SF and whose height (in *y*) corresponded to the front–rear length of the cell. From the original image stack, we extracted an *xyt* substack corresponding to this rectangular region. We then applied ImageJ's Reslice tool to this stack with respect to the *y–t* plane, then used a maximum-intensity projection to collapse the individual slices in *x* to obtain a kymograph in *y* versus *t*. For the kymograph shown in *Figure 4C*, we similarly selected a rectangle the width of a single SF but only 25 pixels high, corresponding to the front end of the SF, and performed a summed projection rather than a maximal projection, to describe the total change in brightness across the width of the SF.

For CK-666 treatments, we dissected egg chambers expressing MRLC-mCh and endogenous Pax-GFP as described above, then separated out older ones leaving only stage nine and younger egg chambers. We placed egg chambers into a final concentration of 1.5 mM CK-666 or the equivalent concentration of DMSO for controls, and either made the slide immediately or after incubating for a period, commencing imaging within a range of 6–70 min.

## Fixed image acquisition and microscopy

We dissected ovaries in live imaging media, as described above, removing muscle sheaths with forceps during dissection to isolate individual ovarioles. We fixed egg chambers for 15 min in 4% EM grade formaldehyde (Polysciences) in PBS (phosphate-buffered saline), and washed them 3× in PBT (PBS +0.1% Triton X-100). To stain with phalloidin, we incubated them in TRITC or Alexa Fluor 488 phalloidin (both 1:200, Sigma), or Alexa Fluor 647 phalloidin (1:100), for 30 min at room temperature or overnight at 4°C, then washed 3× in PBT and mounted them with one drop of SlowFade Antifade (Invitrogen) or Slowfade Diamond Antifade (Invitrogen) onto a slide with a 22 × 50 mm No. 1.5 coverslip. For antibody staining (DAAM-GFP only), we fixed and washed egg chambers as above, and incubated them at 4°C overnight with an anti-GFP-Alexa 488 antibody (1:200, Invitrogen), washed them 3× over 30 min, and mounted them as above.

We imaged tissue using one of two laser-scanning confocal microscopes, either a Zeiss LSM 800 with a ×40 /1.3 NA EC Plan-NEOFLUAR objective or a ×63 /1.4 NA Plan-APOCHROMAT objective running Zen 2.3 Blue acquisition software, or a Zeiss LSM 880 with ×40 /1.3 Plan-APOCHROMAT or ×63 /1.4 NA Plan-APOCHROMAT objective and a Zeiss Airyscan running Zen 2.3 Black acquisition software to improve resolution and signal-to-noise ratio. For all images, a single confocal slice is shown. We did all image processing using ImageJ, as described in the video microscopy section.

For CK-666 treatments, we dissected egg chambers expressing MRLC-mCh and endogenous Pax-GFP as described above, then separated out older ones leaving only stage nine and younger egg chambers. We transferred them to 500 µl live imaging medium containing either 1.5 mM CK-666 in DMSO or the same concentration of DMSO alone and incubated covered for 1 hr at 25 °C before fixing both as described above.

## Measurements from fixed imaging data

We measured the spacing and number of adhesions on SFs from confocal images of endogenously tagged Pax-GFP (DGRC 109-971) costained with Alexa Fluor 647 phalloidin. As depicted in *Figure 3—figure supplement 1*, we selected individual SFs (all those within a given cell that spanned at least half the cell length, and well separated from nearby SFs) using a segmented line, four pixels wide, in the phalloidin channel. We measured the length of this line, then used the Straighten function and the Find Peaks macro in ImageJ (using the default settings of minimum amplitude 44.8 and minimum distance 0) to count number of bright, separate paxillin spots.

We measured SF lateral spacing from images taken in Airyscan mode of stage seven egg chambers stained with phalloidin, containing control and *DAAM*^A mutant mitotic clones. We began by taking a line scan across the middle of the cell just inside the lateral membrane with ImageJ' s Plot Profile tool, and identified individual SFs as having a gray value over 500 (16-bit data) and being at least 0.5 µm from adjoining peaks. Line scans taken at the front and rear of the cells yielded very similar lateral

spacings, indicating this does not vary across the length of the cell. We calculated lateral spacing as the number of SFs occurring across the width of the line scan.

To quantify mean brightness of basal structures in ImageJ, we used single confocal sections of the basal epithelial surface. For relative SF brightness, we used egg chambers mosaic for a DAAM loss of function mutation or Flp-out clones driving RNAi. We employed the irregular polygon tool to manually outline cells, excluding the cell-cell interfaces and leading-edge structures, and measured mean fluorescence intensity of 10 experimental cells and 10 nearby control cells in each egg chamber, and calculated a ratio for each egg chamber. We selected cells in close proximity to avoid possible effects from anterior–posterior gradients along the egg chamber. We employed the same technique to quantify Talin-GFP brightness in egg chambers clonally expressing *DAAM RNAi* or overexpressing activated C-DAAM.

To quantify MRLC brightness, we compared egg chambers expressing *DAAM RNAi* in the entire follicular epithelium to control egg chambers, measuring the majority of the field of cells in view on the slide, but excluding regions near the edge of the egg chamber. We similarly quantified levels of DAAM-GFP (in egg chambers stained with antibody to GFP).

## Quantification and statistical analysis

All data were obtained from at least two independent experiments, and several females were analyzed each time. All data were highly reproducible. No statistical method was used to predetermine sample size. The sample size for each experiment can be found in the figure legend. We tested data for normality using the Shapiro–Wilk test, which is robust to small sample size. In nearly all cases data were normally distributed, but we chose to use nonparametric statistics as more appropriate for the low and uneven sample numbers. We used two-tailed, Wilcoxon matched pairs signed ranks tests, or Mann–Whitney tests, to determine if two datasets were significantly different, with Dunnett's correction when comparing multiple datasets.

Notably, nonparametric statistical tests are less powerful than the corresponding parametric versions, and thus provide a more stringent test of significance. Therefore, we also performed two-tailed ratio paired *t*-tests (*Figure 6D*, *Figure 6—figure supplement 1C, D*, *Figures 7F and 8B*) and unpaired two-tailed *t*-tests (*Figure 8D*) or analysis of variance followed by Dunnett's correction for multiple comparisons (*Figure 6—figure supplement 1*), to ensure that we were not failing to reject the null hypothesis of no effect from genetic manipulations. In no case did the parametric test detect significance not seen by the corresponding nonparametric one.

Analysis was performed using Prism software, version 8 (GraphPad). Experiments were not randomized, nor was the data analysis performed blind. Egg chambers damaged by the dissection process were not included in the analysis.

## Acknowledgements

We thank Hugo Bellen, Nick Brown, Thomas Lecuit, József Mihály, Brooke McCartney, Denise Montell, Eric Wieschaus, and Nir Yakoby for generously sharing reagents. We are also grateful to Yvonne Beckham, Lindsay Lewellyn, and members of the Horne-Badovinac lab for comments on the manuscript.

## Additional information

### Funding

| Funder | Grant reference number | Author |
| --- | --- | --- |
| National Institutes of Health | R01 GM126047 | Sally Horne-Badovinac |
| National Institutes of Health | R01 GM136961 | Sally Horne-Badovinac |
| National Institutes of Health | T32 GM007183 | Maureen Cetera |

| Funder | Grant reference number | Author |
|--------|------------------------|--------|

The funders had no role in study design, data collection and interpretation, or the decision to submit the work for publication.

## Author contributions

Kristin M Sherrard, Conceptualization, Formal analysis, Investigation, Methodology, Validation, Visualization, Writing - original draft, Writing – review and editing; Maureen Cetera, Conceptualization, Methodology, Writing – review and editing; Sally Horne-Badovinac, Conceptualization, Funding acquisition, Project administration, Supervision, Writing - original draft, Writing – review and editing

## Author ORCIDs

Kristin M Sherrard http://orcid.org/0000-0001-6324-2865
Sally Horne-Badovinac http://orcid.org/0000-0002-0473-7451

## Decision letter and Author response

Decision letter https://doi.org/10.7554/eLife.72881.sa1
Author response https://doi.org/10.7554/eLife.72881.sa2

---

# Additional files

## Supplementary files

• Supplementary file 1. Experimental genotypes. Detailed list of the genotype corresponding to each figure panel, also showing the temperature at which females were matured on yeasted food prior to dissection.

• Transparent reporting form

## Data availability

The data in this manuscript include image files, movies, and the quantification of features therein. Numerical data for each graph have been provided as source data excel files tied to each relevant figure. The dataset of images and movies are publicly available at https://doi.org/10.6084/m9.figshare.c.5697073.v1.

The following dataset was generated:

| Author(s) | Year | Dataset title | Dataset URL | Database and Identifier |
|-----------|------|---------------|-------------|------------------------|
| Sherrard K, Horne-Badovinac S | 2021 | Data for DAAM mediates the assembly of long-lived, treadmilling stress fibers in collectively migrating epithelial cells in Drosophila Untitled Collection | https://doi.org/10.6084/m9.figshare.c.5697073.v1 | figshare, 10.6084/m9.figshare.c.5697073.v1 |

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
