## [Editor Report]

Sherrard and colleagues present one of the first examples of in vivo analysis of stress fibers, adhesive structures that are important for cell migration. While stress fibers have been widely studied in cell culture models the relation between these studies and what is actually seen in intact whole animals remains to be established. Using quantitative live imaging and the vast toolkit of genetic tools available in the fly, they characterize a novel treadmilling behavior of stress fibers, providing valuable information about the mechanisms of cell migration and how the collective movement of cells is regulated in vivo.

---

## [Decision Letter]

**Decision letter after peer review:**

Thank you for submitting your article "DAAM mediates the assembly of long-lived, treadmilling stress fibers in collectively migrating epithelial cells in *Drosophila*" for consideration by *eLife*. Your article has been reviewed by 2 peer reviewers, including Derek Applewhite as the Reviewing Editor and Reviewer #1, and the evaluation has been overseen by Anna Akhmanova as the Senior Editor. The following individual involved in review of your submission has agreed to reveal their identity: Guy Tanentzapf (Reviewer #3).

Essential revisions:

1) Figure 3 – Why are the st12 images of adhesion molecules other than Paxillin not shown? Only Paxillin is shown for both st7 and st12, but talin and β-PS are only shown at stage 7. This makes it a little hard to draw general conclusions about the localization of these adhesion molecules at the end of migration. Also, if this is feasible, maybe it would be worth looking into the progression/remodelling of these adhesions from the initiation of migration (not shown in the manuscript) to the migratory stage (shown) to the post-migratory stage (partially shown).

2) Knockdown data: Since some of the formin RNAi knockdown lines might not have great knockdown efficiency, can the authors exclude the contribution of other formins in addition to DAAM? Have the authors tried overexpressing formins, such as DAAM, during and after migration?

3) The kymograph data is beautiful and underlines the treadmilling behaviour, but it is not sufficiently explained. The results are quite obvious from the images in the figure and the materials/methods go into a lot of detail about how to generate and analyze the kymograph, but the corresponding explanations (what is a kymograph? Which question is addressed and what are we supposed to see?) are not entirely clear. A more detailed explanation would be very helpful, especially for readers outside the field.

*Reviewer #1 (Recommendations for the authors):*

Overall I found the manuscript to be very strong and will be of great interest to larger cell migration and cytoskeleton community.

---

## [Author Response]

Essential revisions:1) Figure 3 – Why are the st12 images of adhesion molecules other than Paxillin not shown? Only Paxillin is shown for both st7 and st12, but talin and β-PS are only shown at stage 7. This makes it a little hard to draw general conclusions about the localization of these adhesion molecules at the end of migration. Also, if this is feasible, maybe it would be worth looking into the progression/remodelling of these adhesions from the initiation of migration (not shown in the manuscript) to the migratory stage (shown) to the post-migratory stage (partially shown).

We have added new data to Figure 3, so that all four adhesion markers are now shown both at stage 7 (during migration) and stage 12 (after migration). We agree that it would be interesting to also show the distribution of adhesions before migration starts. However, migration begins right as an egg chamber buds from the germarium. The pre-migratory follicle cells have linear acto-myosin bundles and integrin-based adhesions at their basal surfaces, but the arrangement of these structures is so dense that we cannot tell how they relate to one another. Since the only thing we can conclude is that the pre-migratory cells look different from both the migratory and post-migratory cells, we have not included these data in the revised manuscript.

2) Knockdown data: Since some of the formin RNAi knockdown lines might not have great knockdown efficiency, can the authors exclude the contribution of other formins in addition to DAAM? Have the authors tried overexpressing formins, such as DAAM, during and after migration?

We cannot exclude the possibility that other formins work with DAAM to assemble the stress fibers. In fact, we think it’s likely. Lines 218-220 now read, “From these data, we conclude that DAAM is a key contributor to treadmilling SF assembly. It is important to note, however, that we do not know that all the formin RNAi transgenes we screened are functional, so other formins may work with DAAM in this context”.

We have found that overexpressing full-length DAAM in the follicle cells has no effect on the stress fibers at any stage. However, overexpressing an activated form of DAAM in which the protein is not capable of auto-inhibition (C-DAAM) increases F-actin levels in the stress fibers, as well as Talin levels in the adhesions. In the original manuscript these data were shown for migratory stages in Figures 6C-D and 7F, respectively. In the revised manuscript, we have increased the n value for these experiments, as it was a little low. Over-expressing C-DAAM in post-migratory stages also increases F-actin levels in these stress fibers, which is not surprising given the that the protein is constitutively active. However, this expression causes the stress fibers to have a highly aberrant architecture as shown in Author response image 1. Since it is unclear what we can conclude from these data, we have not included them in the revised manuscript.

**Author response image 1. sa2fig1:** 

3) The kymograph data is beautiful and underlines the treadmilling behaviour, but it is not sufficiently explained. The results are quite obvious from the images in the figure and the materials/methods go into a lot of detail about how to generate and analyze the kymograph, but the corresponding explanations (what is a kymograph? Which question is addressed and what are we supposed to see?) are not entirely clear. A more detailed explanation would be very helpful, especially for readers outside the field.

We have added additional explanation of the kymograph data to both the main text (lines 142-147) and the legend for Figure 4A.